# The Largest Crop Production Shocks: Magnitude, Causes and Frequency

Florian Ulrich Jehn[1,2,3] , James Mulhall[1] , Simon Blouin[1] , Łukasz G. Gajewski[1] , Nico Wunderling[3,4,5]

[1]Alliance to Feed the Earth in Disasters (ALLFED), Lafayette, CO, USA
[2]Societal Dynamics (SoDy), Remote
[3]Center for Critical Computational Studies (C³S), Goethe University Frankfurt, Frankfurt am Main, Germany
[4]Earth Resilience Science Unit, Potsdam Institute for Climate Impact Research (PIK), Member of the Leibniz Association, Potsdam, Germany
[5]High Meadows Environmental Institute, Princeton University, Princeton, NJ, USA

*Correspondence to*: Florian Ulrich Jehn (florian@allfed.info) or Nico Wunderling (wunderling@c3s.uni-frankfurt.de)

**Abstract** Food is the foundation of our society. We often take it for granted, but stocks are rarely available for longer than a year, and food production can be disrupted by catastrophic events, both locally and globally. To highlight such major risks to the food system, we analyzed FAO crop production data from 1961 to 2023 to find the largest crop production shock for every country and identify its causes. We show that large crop production shocks regularly happen in all countries. This is most often driven by climate (especially droughts), but disruptions by other causes like economic disruptions, environmental hazards (especially storms) and conflict also occur regularly. The global mean of largest country-level shocks averaged -29%, with African countries experiencing the most extreme collapses (-80% in Botswana), while Asian and Central European nations faced more moderate largest shocks (-5 to -15%). While global shocks above 5% are rare (occurring once in 63 years), continent-level shocks of this magnitude happen every 1.8 years on average. These results show that large disruptions to our food system frequently happen on a local to regional scale and can plausibly happen on a global scale as well. We therefore argue that more preparation and planning are needed to avoid such global disruptions to food production.

## 1 Introduction

Having enough food available is essential for every society. However, no food is storable forever, and storage is expensive. As a result, there is always only a very finite amount of food in stock. If production were to stop tomorrow, stocks globally would only last just under a year, with Africa and parts of Asia having only around six months of food stored (Laio et al., 2016). Some important staple crops like wheat would even be depleted in two to three months if production ceased in the months of low stocks and consumption stayed constant (Do et al., 2010). Over the last few decades, however, there has been a trend towards maintaining somewhat larger food stocks, increasing resilience (Laio et al., 2016; Marchand et al., 2016).

One safeguard against the depletion of stocks is the global and interconnected food production and trade system that has developed since the mid-20th century (Ji et al., 2024). In the last few decades, this system has been quite successful in ensuring food security for a majority of the world (Herre et al., 2017). However, in such complex and connected systems, there is always the potential for cascading failures, starting from one local shock and rippling outwards (Bernard de Raymond et al., 2021). Also, the system is highly concentrated among a few key players, like Russia for wheat, the United States for maize, or Brazil for soy. This concentration of food production has historical roots. As Clapp (2023) demonstrates, capitalism and colonialism drove specialization in single crops for efficiency and profitability, while also promoting the distribution of the production system globally, whereby certain regions or countries specialize in producing certain types of goods — grains, fruits, textiles, etc. This required these countries to then become bulk importers of the goods they did not produce themselves. Post-colonial countries inherited economies dependent on food imports rather than local production. This has created a system where disruptions to a few key crops or exporting nations can have cascading global effects, with recent research by Jain (2024) showing that this concentration also happens on a country level, with certain regions in a given country being responsible for most of the production and trade.

There have been a variety of studies to understand the events that might cause such an abrupt loss in food production. One of the more comprehensive examples is Cottrell et al. (2019), who looked at food production shocks across crops, livestock, fisheries, and aquaculture and found that the frequency of shocks increases over time, and that the shocks are mainly caused by climate and geopolitical disruptions. Another way to analyze these global shocks is the concept of Multiple Breadbasket Failure (MBBF). This term describes the dangers that arise in the food system when several of the main food-producing regions globally experience a yield shock in parallel (Gaupp et al., 2020; Jahn, 2021).

More recently, a new term has been introduced for another kind of risk to the food system: Global Catastrophic Food Failure (GCFF) (Wescombe et al., 2025). This term is meant to describe the gravest risks our food system could face, disruptions so large that food production would exhaust stocks and lead to widespread famine if not managed well, due to e.g. climate change, war, volcanoes, or pandemics. A shock of this magnitude entails a significant risk of creating famine on a large scale.

Such grave shocks have not happened since data collection by the FAO started in 1961. For the time before this, data only exist for a small subset of countries (Anderson et al., 2023), so it is considerably more uncertain to what extent food production shocks occurred before that. The most plausible events that might have caused such a global shock in the last century were the two world wars, but data from that period are patchy. Another historical candidate for a GCFF is the eruption of Mount Tambora in 1815 and its climatic consequences, but the records of yields from that time are too sparse to be certain (Brönnimann and Krämer, 2016). Unfortunately, our modern food system is vulnerable to disruptions on global scales by events like nuclear war (Xia et al., 2022), geomagnetic storms or extreme pandemics (Moersdorf et al., 2024) and large volcanic

eruptions (Cassidy and Mani, 2022). Also, as we further move towards polycrisis, it becomes more likely that several shocks coincide at the same time (Delannoy et al., 2025).

Such extreme risks often seem abstract and distant, making them seem implausible. To address this perception gap, this paper aims to ground future catastrophic food security risks in historical data. To do so, we aggregate all major crops based on their caloric value to have an overall measure of food production. We focus on crops because they make up the majority of calories consumed by humans (>85%), and there is very reliable data available. We aggregate the crops by calories because, without enough calories, you cannot prevent famine. This provides us with a time series (1961 to 2023) of calories produced for all countries, from which we can calculate how much the actual yield differs from the expected yield based on long-term trends in food production.

Our approach builds on previous work, such as Cottrell et al. (2019) and Anderson et al. (2023). However, rather than analyzing climate patterns that might cause shocks like Anderson et al. (2023) or identifying shocks across multiple food sectors like Cottrell et al. (2019), this paper systematically describes the worst crop production shock that each country experienced and why it happened. We believe this unique focus on the largest magnitude shocks highlights the greatest dangers that crop production faces, providing a comprehensive map of actual worst-case vulnerabilities rather than merely describing risk factors in general. This study here complements Cottrell et al. (2019). While the earlier study focussed on how often the food system shows shocks in general, this study here explicitly focuses on how bad these shocks can get and why these most extreme shocks happen.

Our comprehensive shock dataset enables investigation of three key research objectives. First, we aim to quantify the magnitude of the most severe crop production shocks to establish baseline thresholds for extreme events. Second, we aim to analyze temporal trends in the frequency of largest shocks to identify whether extreme events are becoming more or less common over time. Third, we aim to identify and categorize the primary drivers of these production shocks to understand their underlying mechanisms.

## 2 Data and Methods

### 2.1 Data

To conduct our analysis, we used food production data provided by the Food and Agriculture Organization of the United Nations (FAO). This dataset covers all major crops and contains data from 1961 to 2023. We used the main crops in each of the main crop types as described by FAO (2024):
- Cereals: Maize, rice, wheat, barley, sorghum
- Sugar crops: Sugar cane, sugar beet

95          ● Vegetables: Tomatoes, onions (including shallots), cucumbers and gherkins, cabbages, eggplants

96          ● Oilcrops: Oil palm fruit, soya beans, rapeseed, seed cotton, coconuts

97          ● Fruit: Bananas, watermelons, apples, grapes, oranges

98          ● Roots and tubers: Potatoes, cassava, sweet potatoes, yams, taro


Using all these crops means we are considering the vast majority of crops produced globally. We aggregate all of these crops
based on their caloric value. To stay consistent with FAO data, we also use FAO caloric density estimates (FAO, 2001a). To
get the overall caloric production, we multiply the production values of the foods by their calories and sum all calories produced
in a given year and country.

We do not differentiate between which of these crops are intended for feed or food, because in a famine situation, we assume
that most, if not all, of it would be used for human consumption. We recognize that this does not reflect current food
consumption patterns, because several of the crops (like maize or soya beans) are mostly used for feed and only 55% of global
crop calories reach humans directly  (Cassidy et al., 2013). However, our aim is to quantify crop production shocks, rather
than current consumption patterns. During severe food crises, feed is often redirected towards human consumption. For
example, there are documented cases of this phenomenon for both World Wars (Collingham, 2012; Offer, 1991) and during
the Great Chinese Famine (Meng et al., 2015). Depending on the crop, this might take some time and infrastructure, but it
represents a sensible crisis response. Most of the crops we consider here are directly edible by humans. The crops used here,
which are likely the most difficult for humans to consume, are seed cotton, rapeseed, and soya beans. To assess whether this
changes our findings, we redid the analysis excluding seed cotton, rapeseed and soya beans. The results stay almost exactly
the same, and for most countries, the results only change by a percentage point or less. This can also be seen in Figure S1,
which is a version of Figure 2 but without those crops. The changes are so small that they are almost not detectable visually.
We therefore conduct the analysis with the whole set of crops.
**2.2 Calculating food shocks**
For this analysis, we consider it a food shock if the amount of crops produced in a given year is considerably lower than the
amount of crops we would expect for that year. However, to calculate this shock, we must first estimate the expected yield for
that year. To do so, we are using a Savitzky-Golay filter (Savitzky and Golay, 1964) as implemented in scipy v1.15.2 (Virtanen
et al., 2020).
The Savitzky-Golay filter is a smoothing technique that reduces noise in data while preserving important features like peaks
and trends. It works by fitting a polynomial to small subsets (a window) of neighboring data points, then using the polynomial
to estimate a smoothed value at the center of each subset. At each position, the filter fits the best polynomial curve through the

data points within that window, then takes the value of that curve at the center point as the smoothed result. This process continues across the entire dataset.

This process is similar to the food shock calculation in Anderson et al. (2023), who used a Gaussian filter. We chose the Savitzky-Golay filter because it performs better at the edges of the dataset. We use a window length of 15. This means the 7 years before and after a given year are used to calculate the expected value for that year. We used this window length to make our approach comparable to Cottrell et al. (2019). Cottrell et al. (2019) considered in their shock calculation the previous 7 years. We used a 3rd order polynomial, as this resulted in an overall smoother estimation. Though ultimately, a Gaussian filter and the Savitzky-Golay filter deliver very similar results for our dataset and identify similar magnitudes of shocks, as well as the same years with the largest shocks (Figure S2).

For the detection of the largest shocks, we also introduced a conditional constraint. We only count a relative drop in crop production as a shock if the crop production in the shock year is lower than the previous year. This is to avoid detecting a year as having a shock, even though the amount of food produced has increased, which can happen if there is a sudden increase in production in the following years. The additional constraint was added because the initial analysis incorrectly flagged years as shocks when yields had actually increased from the previous year. However, having more crops than the year before can hardly be considered a shock.

However, our overall analysis is relatively robust against changes in the window size and polyorder, as the overall trend follows a relatively smooth curve to begin with (see Figure 1 for an example). Smaller windows decrease shock sizes because the smoothed trend follows the yearly data more closely. Larger window sizes lead to larger shock sizes accordingly. The overall trends remain very similar because the positions for potential large derivations do not change, even if the individual shock sizes do. See Figure S3 and S4 for a re-calculation of Figure 1, but with 7 and 21 years for the calculation of the trend line. This shows that the values slightly change, but in all three cases it highlights the same three years, in the same order, as the largest shocks in the time series.

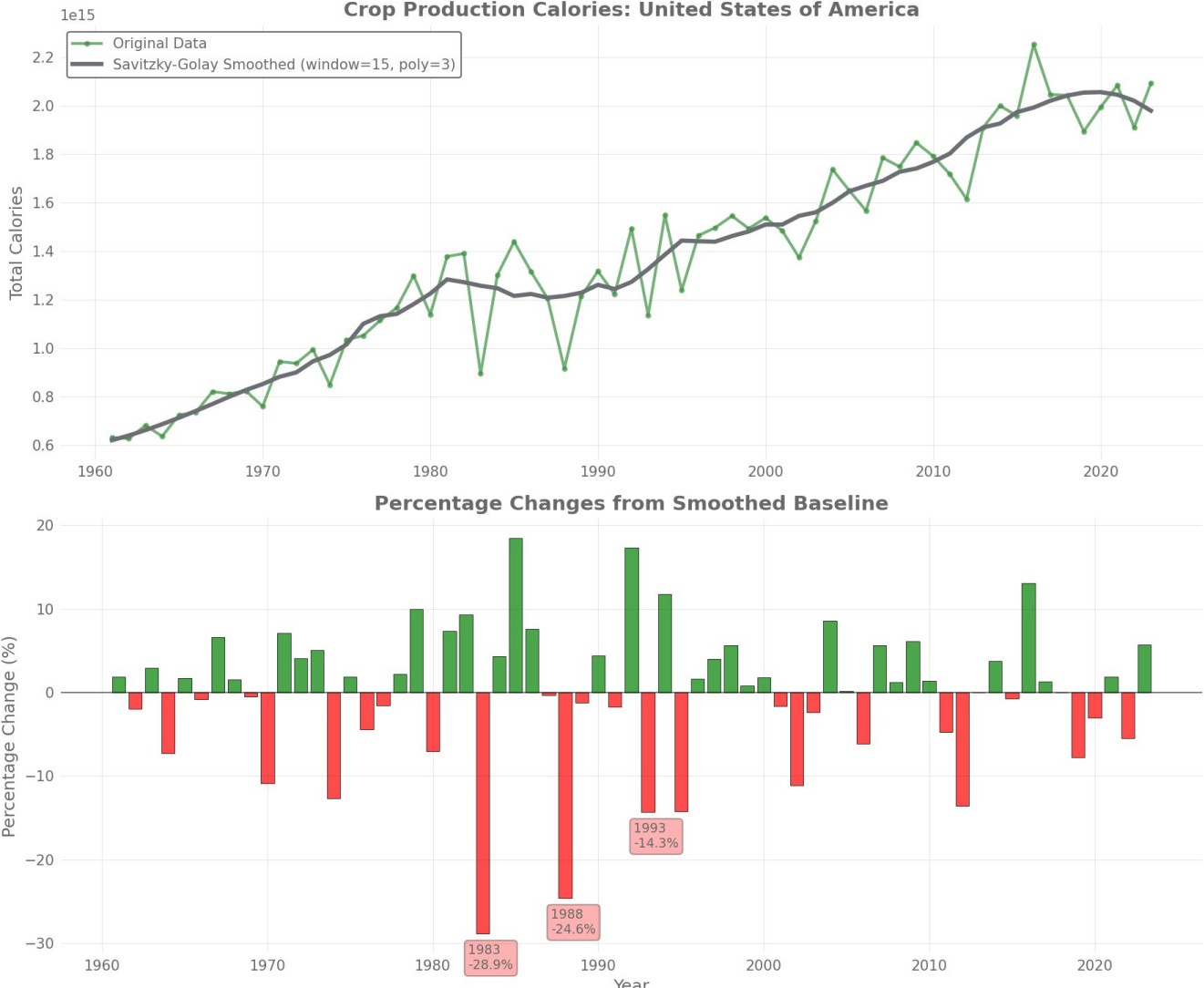

**Figure 1: Example of crop calorie production in the United States (1961-2023). Upper plot shows original calorie data in green and smoothed trendline calculated with Savitzky-Golay filter in grey. The lower plot shows the size of the crop production shock calculated with our method. Green represents more calories produced than expected, red represents less calories produced than expected. The three largest shocks are labelled with the year and size of the shock.**

We used both historical and contemporary countries, which slightly inflates shock counts when borders changed—a shock affecting one territory before partition now registers across multiple successor states (e.g. the Soviet Union and its successor states). However, this effect is negligible, and the number of countries stabilized around 1990.

For this analysis, we considered a total of 197 countries. We did not exclude countries with small crop production, as there is
no clear cut-off point, and exclusion would have been arbitrary. However, for these small countries, it is more difficult to
explore reasons for their crop production shocks, as there is less documentation available, and smaller production numbers are
more easily skewed.
**2.3 Checking the origins of the largest food shocks**
To verify if our approach reliably finds the largest food shocks in a country's history, we used Claude 4 Sonnet to search for
potential crises in these countries that might have caused the food shocks we had detected (full prompt can be found in the
repository of this paper). This provided us with several official sources (e.g. journal articles, FAO reports) that described a
crisis in a given year and country. Each search result was verified manually reading through the sources suggested by Claude
and confirming whether they described a crisis in the specified country and year specified that could have influenced
agriculture on such a scale. While this might produce some false positive results, it is also an approach used by Cottrell et al.
(2019) and the magnitude of the events identified fits with the size of the shocks.
The way this search was conducted means Claude was only used to find sources to verify with, but the actual verification was
done by humans with independent sources, avoiding the danger of hallucinations and related problems in large language
models. If no reasonable source was provided by Claude, we searched for the reason with a normal internet search. If this also
did not bring up anything plausible, we sorted this shock into the "Unknown" category. The reason for the shock had to occur
in the year of the shock or the year before to be counted. If reliable sources were found, we used those to classify the shock
into one of the following categories:
● **Conflict** - wars, civil unrest, territorial disputes
● **Economic** - financial crises, currency devaluation, market collapse
● **Climate** - droughts, extreme temperatures, late cold spells
● **Pest/Disease** - crop diseases, locust invasions, livestock epidemics
● **Policy** - agricultural policy changes, land reforms, trade restrictions
● **Mismanagement** - soil degradation, overexploitation, poor planning
● **Environmental Hazard** - storms, tsunamis, earthquakes, volcanoes
● **Unknown** - insufficient information found
We used these categories, following the approach in Cottrell et al. (2019), but disaggregated some of them to get more fine-
grained results. This process allowed us to assign a crisis to almost all of the shocks we detected. Also, many of the sources
we used to verify the shocks used phrases like "worst drought year... since the mid-15th century"; Tunisia in 2002 (Ghoneim
et al., 2017), "most violent and bloody period of the entire armed confrontation"; Guatemala in 1984 (HRDAG, 1999) or
"driest hydrological year on record"; Greece in 1977 (Vasiliades and Tzabiras, 2007). This suggests that our method is able to
detect the worst shock to have occurred in these countries.
We categorized shocks by their primary driver while recognizing that most agricultural crises involve multiple interacting
factors. Our classification captures the dominant cause that initiated or most directly drove the production decline. For example,
while economic factors often compound climate shocks, we classified droughts as 'climate' when reduced rainfall was the
primary trigger, even if currency devaluation worsened the impact. This approach provides clarity about initial drivers while
necessarily simplifying complex causal chains. The 'shock' timeframe in our analysis is annual, based on year-to-year
production changes. Multi-year cascading effects—where one year's climate shock leads to mismanagement that causes
another shock—are captured as separate events in our dataset.
For some countries where we could not identify a clear cause, the food shocks were either minor or occurred in nations with
low crop production. In these cases, even small absolute declines appeared as major shocks (e.g. Puerto Rico). Additionally,
some countries showed data patterns like maintaining low production for decades, then experiencing sudden jumps that
increased food production by an order of magnitude from one year to the next, with production remaining at this higher level
afterwards (e.g. Oman). These patterns suggest problems with the country-level data rather than flaws in our methodology.
The list of the largest food shocks for each country can be found in the repository and in the supplementary materials as a
comma-separated values (CSV) file, complete with yield change, year, category, reason, and source.

## 2.4 Calculating global correlations

In order to investigate the relationships between country-level shocks and global shocks, we calculated the Spearman
correlation coefficient between each country and the rest of the world. This was done to see which countries experience changes
in food production similar to global patterns, and which countries deviate. We chose Spearman over other correlation
coefficients, such as Pearson, because we are interested in whether there is a monotonic relationship between countries (e.g.,
whether countries experience shocks or surpluses at the same time), but not whether this relationship is linear. The rank-based
nature of Spearman correlation also makes it robust to outliers and prevents countries with large production magnitudes from
disproportionately influencing the correlation. The process was done for each country by subtracting the annual crop
production of that country from the world crop production, applying the Savitzky-Golay filter as described in Section 2.2 to
calculate the yield changes for the world minus that country, and then calculating the correlation. This was done to avoid
spurious correlations, since each country's production would otherwise be part of the global numbers.

## 3. Results

### 3.1 Magnitude of crop production shocks

The magnitude of the largest crop production shocks varies considerably across countries (Figure 2). Africa stands out with several nations experiencing extreme production collapses—Zimbabwe reached -70% in 1992, while other Southern African countries show similarly severe declines exceeding -70%. This geographic concentration of extreme shocks in Southern Africa suggests regional vulnerability to shared climatic or economic disruptions. North Africa and parts of the Middle East also display substantial shocks ranging from -40% to -60%, indicating widespread agricultural vulnerability across the continent. By contrast, countries in Asia and Central Europe typically face more moderate shocks (-5 to -15%), with this being seen in Southeast Asian nations in particular. This pattern partly reflects the temporal scope of our analysis—China, for instance, experienced major crop failures shortly before the FAO dataset began in 1961 (Meng et al., 2015).

The majority of countries fall between these extremes, with the global mean of the largest shocks averaging approximately -29%. South America presents an interesting case of relatively mild maximum shocks across most of the continent.

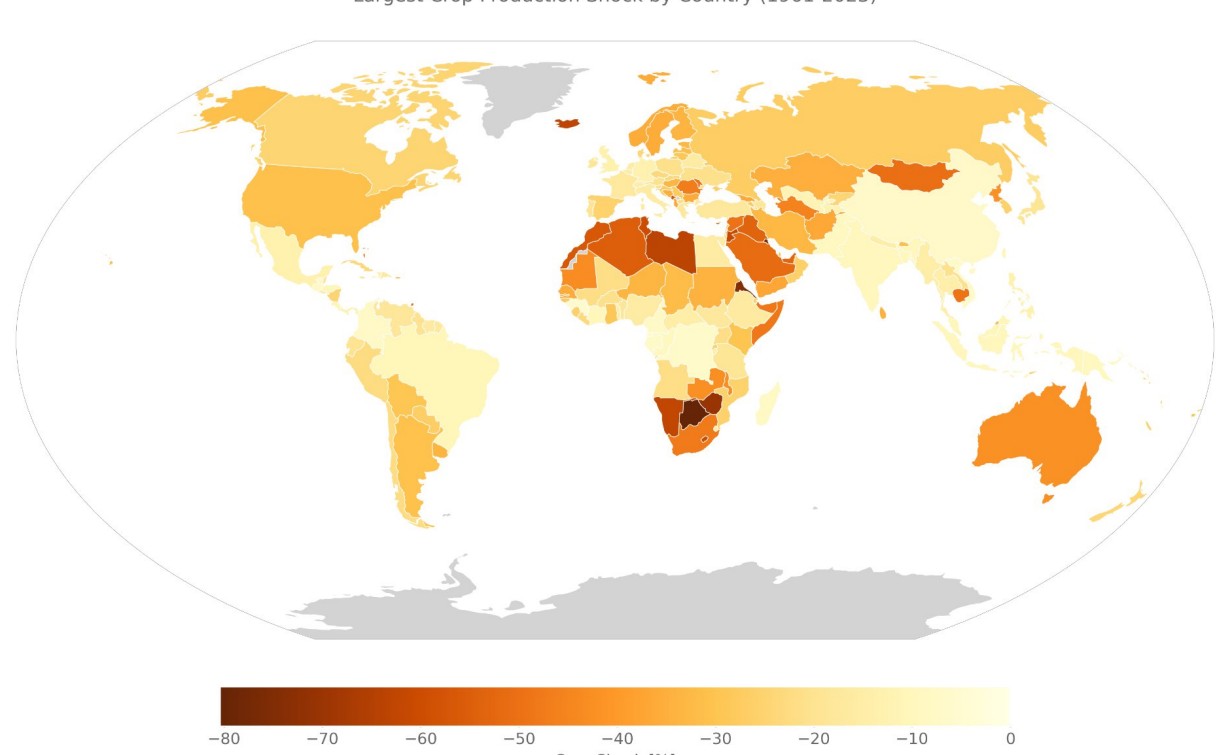

Largest Crop Production Shock by Country (1961-2023)

Figure 2: Largest crop production shock in all present-day countries. Darker colors indicate larger shocks. Grey indicates no data. The shocks are calculated as deviations from expected yield in a given year and are based on the combined calories from all assessed crops.

However, the largest crop production shocks differ not only in their geographic distribution, but their magnitude also varies substantially depending on the underlying cause (Figure 3). Climate-related shocks demonstrate the most severe impacts, with a mean around -32% but extreme cases reaching -80%—predominantly driven by droughts. This category shows the widest distribution of impact severity, reflecting the diverse nature of climate hazards, from moderate seasonal variations to catastrophic multi-year droughts.

Human-caused shocks generally result in smaller production declines and show more constrained distributions. Policy interventions produce the least severe impacts (mean -21%), while economic disruptions show similar severity (mean -21%). Mismanagement displays a mean of -30% with a relatively tight distribution. Conflict presents moderate average impacts (mean -27%) but high variability, from minor disruptions to catastrophic losses exceeding -70%.

Environmental hazards occupy a middle position with a mean of -29%, though their distribution is more concentrated between -10% and -40%, primarily caused by tropical storms. The "Unknown" category shows substantial variability (mean -27%), likely reflecting the diverse mix of unidentified shock types.

The distinction between natural and human causes becomes increasingly blurred as anthropogenic climate change intensifies both drought frequency and tropical storm severity. Having only one data point for pests and diseases makes it difficult to compare to the other categories, as it could just be a random occurrence. However, as it is smaller than almost any other data point implies that pests and diseases are not a major factor for the largest shocks. This is likely due to pests and diseases often being specific to a single crop, while we looked at a large aggregation of crops. Pests and diseases are often one of the largest sources of crop lossesl (Savary et al., 2019). However, given that we do not find them here as one of the main causes of the largest shocks, this implies that they cause damage on a high magnitude but without large fluctuations.

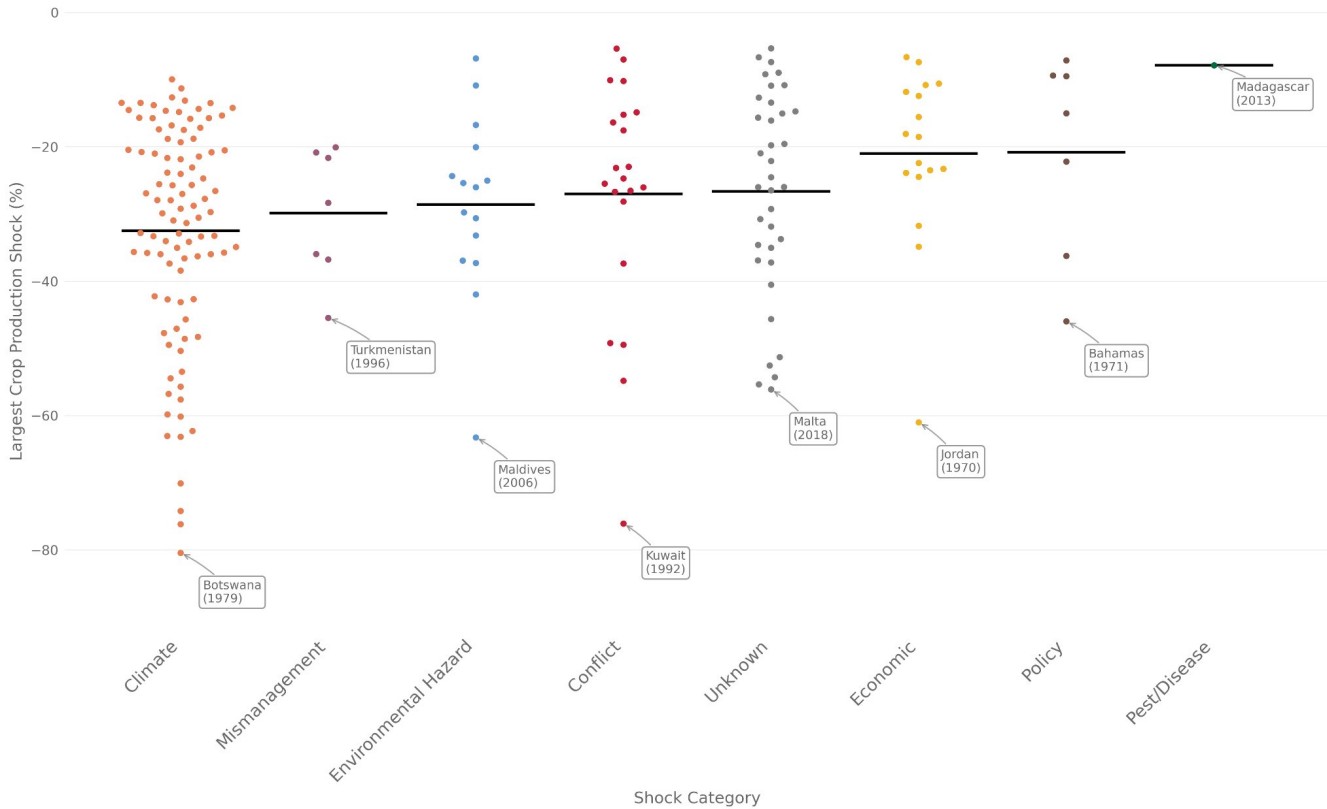

249

**Figure 3: Swarm plots showing the magnitude of crop production shocks across different cause categories. The black line indicates the mean. Single points show all individual country-level shocks. For each category the largest shock is labelled with year and country it occurred in.**

## 3.2 Geographic patterns of shock types

Crop production shocks show clear spatial patterns across continents, with distinct regional concentrations of different shock types (Figures 4, 5). While most shock causes appear on all continents, certain drivers cluster more heavily in specific regions.

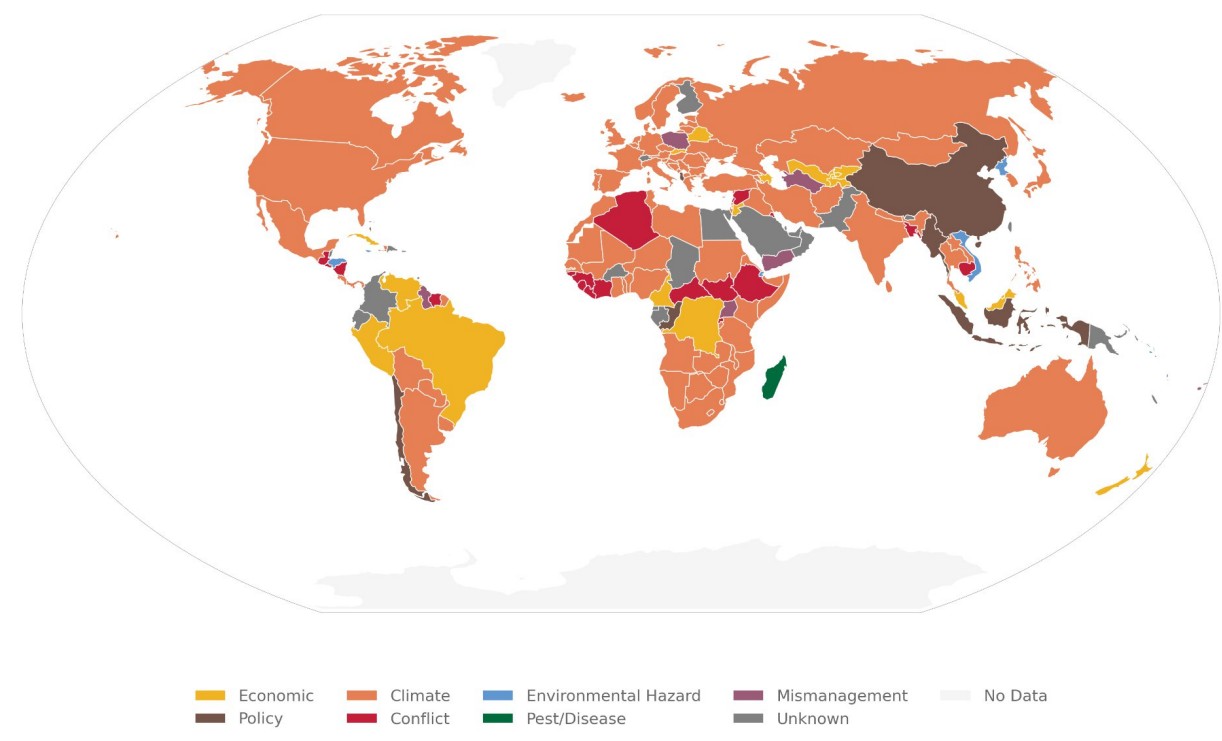

Reason for Largest Crop Production Shock by Country (1961-2023)

Economic    Climate    Environmental Hazard    Mismanagement    No Data
Policy      Conflict   Pest/Disease            Unknown

**Figure 4: Global map showing the main reason the largest crop production shock in a given country happened.**

Europe is quite homogenous; climate shocks dominate almost entirely, comprising roughly 70% of all major production disruptions. Most trace back to the devastating 2003 heat wave that brought extreme temperatures across nearly the entire continent (IPCC, 2007). The few exceptions reveal Europe's otherwise stable agricultural systems: Poland's failed agricultural reform in 1980 (Mandel, 1982), or Belarus facing spillover from Russia's 1999 financial crisis (FAO, 1999).

In North America all the continent's major economies experienced their largest shocks from droughts—Canada in 2002 (Wheaton et al., 2008), the United States in 1983 (Zipper et al., 2016), and Mexico in 1979 (Simons, 1980) (Figure 4). In Central America small Caribbean nations are mostly affected by substantial environmental hazard impacts like tropical storms (Figure 5).

South America shows the highest proportion of economic disruptions among all continents. Brazil faced severe disruption in 1978 from high debt and inflation following oil shocks (Vellutini, 1987), Peru suffered hyperinflation in 1992 due to failed policies and debt burdens (Velazco, 1999), and Venezuela's 1976 focus on oil production came at agriculture's expense (Smith, 2019). Policy-driven shocks are also present, for example Chile's 1973 land reform disrupted production systems (U.S. Central Intelligence Agency, 1972). Conflict appeared in Suriname's 1990 civil war (Reuters, 1991), while climate shocks hit Argentina

in 2009 (Sgroi et al., 2021), Bolivia in 1983 (UN Department of Humanitarian Affairs, 1983), Uruguay in 2018 (Weather Underground, 2018), and Paraguay in 2012 (USDA Foreign Agricultural Service, 2012), all due to drought.

Africa also shows a diverse shock distribution, with conflict driving more production disruptions than any other continent. Civil wars devastated agriculture in Algeria in 1994 (Martinez, 2000), military coups and violence disrupted Guinea in 2009 (UNDP, 2023), and Rwanda's 1994 genocide destroyed agricultural systems as well (FAO, 1996). Despite this conflict prevalence, Africa also experiences all other shock types. Madagascar's 2013 locust swarms destroyed crops across vast areas (FAO, 2013), Cameroon's 1987 economic crisis rippled through agriculture (Tambi, 2015), Djibouti was hit by massive floods in 1989 (UN Department of Humanitarian Affairs, 1989), Congo's 1991 democratization and switch from a more socialist system likely led to disruption in agriculture (IFES - The International Foundation for Electoral Systems, 1992), and Uganda faced the agricultural consequences of nearly a decade of mismanagement under Idi Amin, ending in 1979 (Honey and Ottaway, 1979). Nevertheless, climate—especially drought—remains the primary shock driver, as across all continents.

Asia's shock distribution resembles Africa's, but with fewer conflicts and more economic crises. Conflicts that did disrupt production include Cambodia's 1974 civil war (Defalco, 2014), worsened by US bombing campaigns, and Bangladesh's 1972 post-independence aftermath (Dowlah, 2006). Policy changes created major disruptions when China shifted agricultural support policies in 2003 (Yu et al., 2018) and Myanmar nationalized rice production in 1966 (Steinberg, 2019). Environmental hazards struck repeatedly—North Korea faced devastating floods in 1996 for the second consecutive year (FAO, 1997), while Vietnam endured severe storms in 1978 (Cima and Library of Congress, 1989). As elsewhere, drought-driven climate shocks dominated, exemplified by India's massive 1987 drought (FAO, 2001b).

Oceania's shock patterns prove difficult to assess due to high proportions of unknown causes, likely reflecting both the region's many small island states and limited data availability. Small agricultural sectors trigger shock detection more frequently, while these nations' limited resources and global attention make information gathering challenging. Where causes are known, climate events and environmental hazards—particularly storms—dominate the region's agricultural disruptions.

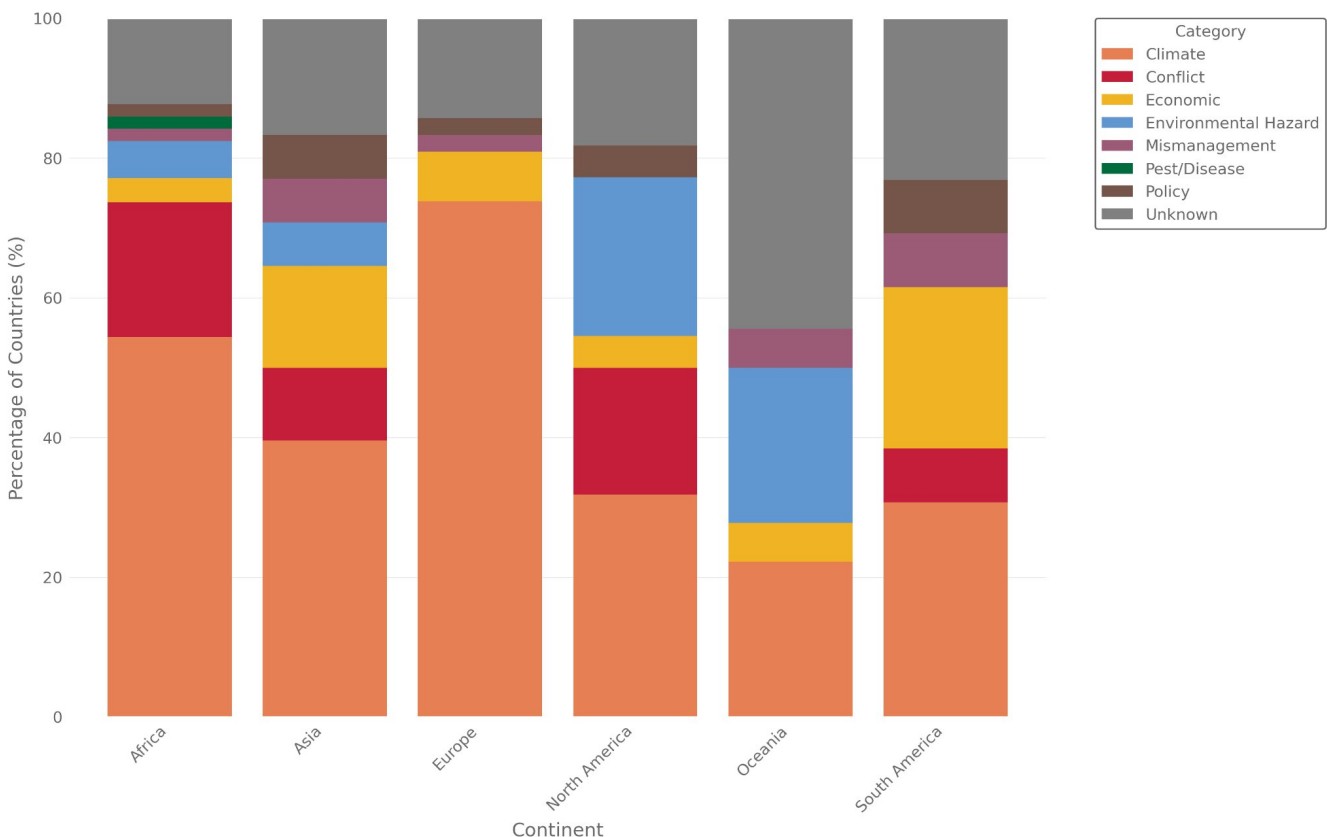

293

**Figure 5: Relative distribution of the main reasons why the largest crop production shocks happened in a country separated by continent.**

### 3.3 Temporal evolution and frequency distribution

When it comes to the temporal evolution of the largest food shock, we can see some clear patterns (Figure 6). All decades except the 1960s and 2020s have a roughly similar number of shocks. This number is also shaped by how many countries existed at a given point in time, but even when we correct for the number of countries that existed in that decade, the 1970s to 2010s all have 15-25% of the countries that existed experiencing their largest shock in that decade (Figure S5). This means the pattern here remains roughly the same, independent of the number of countries which existed.

The pattern that the first and last decades show a small number of shocks seems to imply that our method is less able to detect shocks at the edges of the time series. However, this effect does not happen if we only use the 1970s to 2010s in our analysis (Figure S6), indicating that this is an actual trend in the data and that, especially the 2020s, have had a surprisingly small number of very large crop shocks. Given the base rate over the other decades, this implies that we can expect many more large crop production shocks in the rest of the decade.

The reasons for those largest shocks show that climate-caused crop shocks make up a much larger percentage of cases in the more recent decades. Climate-related shocks grew from about 25% in the 1960s to 50-60% by the 2000s-2010s. This increase corresponds with decreases in other categories, including mismanagement and policy failures. Conflict and unknown causes stay on a similar level throughout, while all other categories tend to become less common over time. The levels of shocks which could not be attributed to a specific cause are at a similar level as in Cottrell et al. (2019).

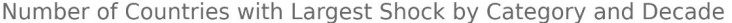

Number of Countries with Largest Shock by Category and Decade

**Figure 6: Absolute distribution of the main reasons why the largest crop production shocks happened in a country, separated by the decade they occurred in. The overall size of a bar indicates the total amount of the largest shocks for a given country in a given decade. Note that the last bar only consists of the four years 2020-2023 and not the whole decade like the other bars.**

We can also look at how the general frequency of the crop shocks varies over the whole time series (Figure 7). This is for shocks on a global level. We can see that crop production shocks happen on a variety of levels, but on a global scale, the largest was just over 5.5%. This was in 1988, mainly caused by a severe and widespread drought in the USA. In this year, the production in the USA declined by 29%, while the USA produced around 20% of all crop calories globally. This highlights how the whole food system can be affected by shocks in even a single country. The distribution shows a sharp decline in

frequency as shock size increases - small shocks of 0-1% happen about 48% of the time, while shocks over 3% occur only 10% of the time, and those exceeding 5% are rare at less than 2%.

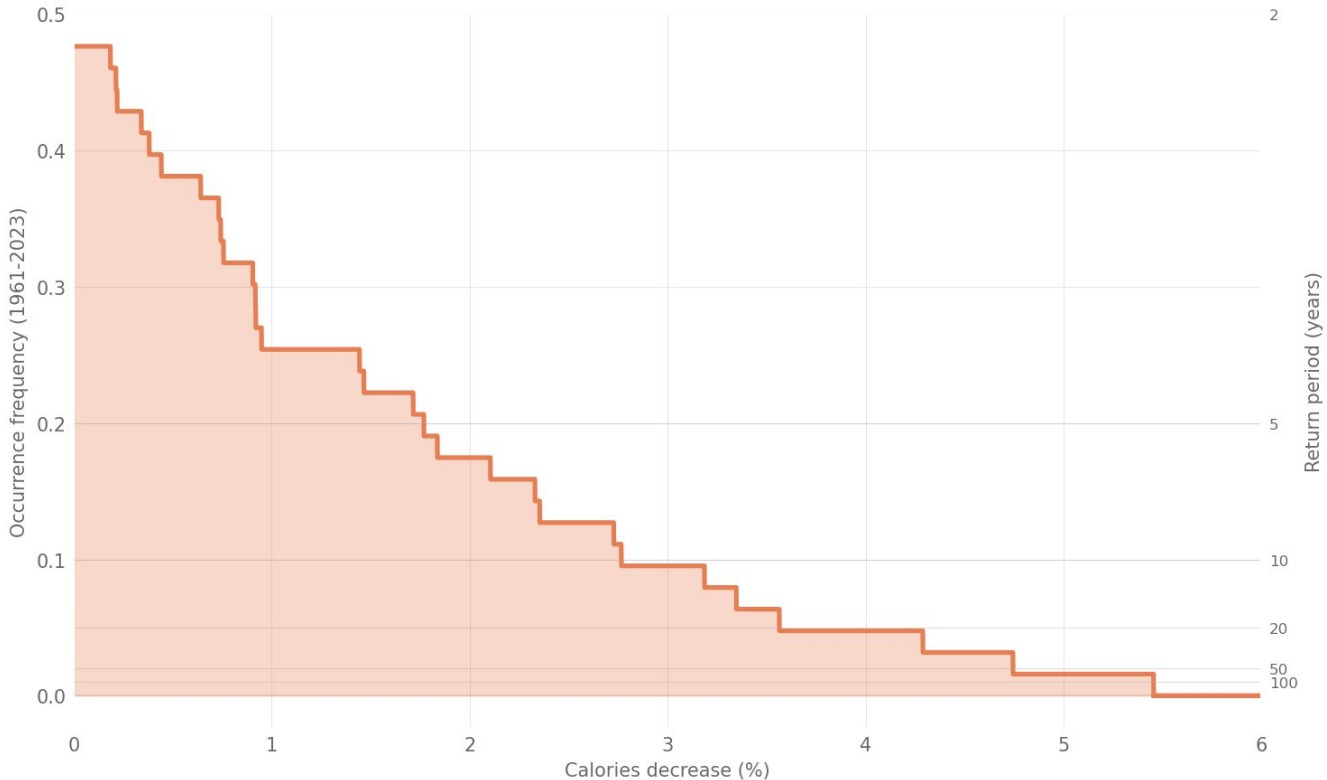

**Figure 7: Frequency of global-level shocks to overall caloric production. The plot shows how often values of losses are exceeded. For example, shocks of 3% or more have been happening 10% of the time. The second y-axis shows the return period of shocks for a given size.**

While shocks exceeding 5% are rare at the global level, occurring only once in our 63-year dataset, they are much more common at the continent and country level. There were 51 continent-level shocks of 5% or more between 1961 and 2023, with at least one happening every 1.8 years on average. At the country level, shocks over 5% occurred every single year, amounting to a total of 2800 shocks.

**3.4 Global synchronization**

From the previous sections, we know that large shocks regularly happen, but also that they usually cancel each other out on a global level. To understand how this manifests, we also looked at how country-level crop production correlates with global

crop production (Figure 8). This shows that there are two groups of countries with opposing production patterns. One group tends to have high crop production when global production is high and low production when global production is low. The other group shows the reverse pattern—low production when global production is high and high production when global production is low. The globally asynchronous countries are most of Africa, parts of the Middle East, Central Asia and the northern part of South America. The synchronous countries are everyone else. European countries are especially synchronous with global production. This is likely due to Europe being a major contributor to global crop production, but not having a large spatial extent. Due to this, if there is a drought in Europe (as for example in 2003), most European countries are affected and thus also global production to a large extent. The asynchronous countries also all share tropical and subtropical climate zones.

The high synchrony observed across North America, Europe, and major Asian producers like China and India suggests these regions respond similarly to large-scale climate phenomena such as El Niño/La Niña events. This synchrony, while contributing to global production stability under normal conditions, also implies that extreme events affecting these regions simultaneously could pose significant risks to global food security. The asynchronous regions, despite often having less stable individual production, therefore play an important buffering role in the global food system by providing production when major producing regions experience shortfalls.

In addition, many of the most asynchronous countries (like Brazil, Ethiopia, or Syria) have conflict and economic reasons for their largest crop production shock. This suggests that the asynchronicity might also be due to those countries being disrupted by internal problems, while the rest of the world did not have these problems on such a scale.

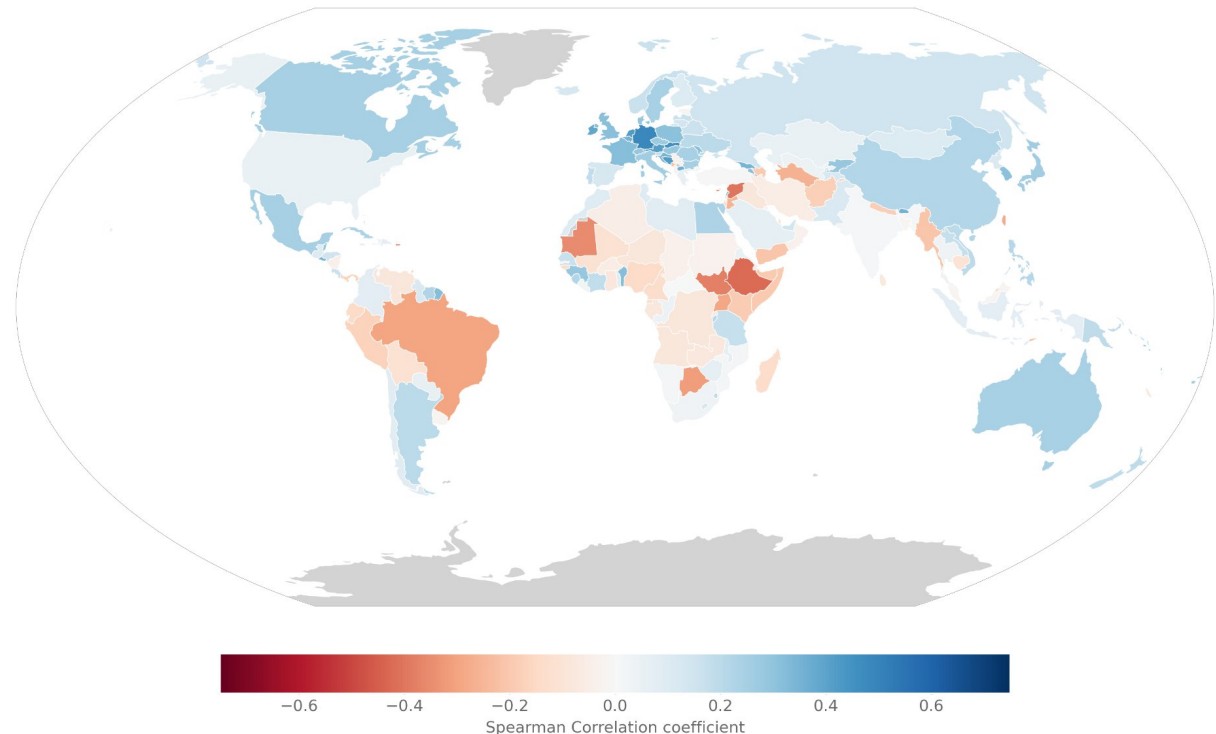

Correlation of crop production changes between each country and the rest of the world

**Figure 8: Correlation of crop production changes between each country and the rest of the world. A positive correlation (blue) means a country's crop production tends to move in the same direction as global production. A negative correlation (red) means the country's production tends to move opposite to global trends.**

These patterns of synchronization mostly stay consistent over time, but there are changes. For example, if we only look at the data from 2003 to 2023 (Figure S7), we can see that Brazil's relatively strong negative correlation becomes positive, while Europe's generally positive correlation and Africa's generally negative correlation persist. This likely reflects changes in agricultural practices and the dominance of certain regions when it comes to crop production.

Much of the synchronicity in global crop production is driven by wheat, which makes up a substantial share of total output (Figure S8). This means wheat's year-to-year variation can overshadow more localized patterns in other crops. For our purposes, this is not a limitation—we aim to understand global crop production as a whole and identify which regions might serve as buffers when others fail. The wheat-driven pattern shifts dominance toward major wheat exporters like Russia and Ukraine, whose production swings carry outsized weight in global totals. The United States, despite being a major wheat producer, shows weaker correlation with global trends. Possibly because its production variability operates independently of the factors driving Eurasian wheat yields, as both continents experience different climate impacts.

## 4. Discussion and conclusion

### 4.1 Climate is the main reason for the worst crop production shocks

Our results clearly show that climate is most often the reason for the largest crop production shock in a given country. This is mostly due to drought, but there are also instances of early frosts and torrential rain. This is concerning, as climate is not only the most common reason for the largest crop production shocks, but it also seems to be increasing over time, likely due to climate change making extreme weather, especially droughts and heatwaves, more likely (Fanzo et al., 2025; Grant et al., 2025). Potentially, climate might also be increasing as a cause as other reasons are getting managed better. For example, conflicts have been decreasing from a peak in the 1980s until around 2010 (Szayna et al., 2017), but have seen a steep uptick since then (Davies et al., 2023). Similarly, since the 1990s, after the fall of the Soviet Union, it might be the case that there are fewer policy and economic-caused crop failures, because most of the world is organized under neoliberal capitalism and no new approaches to organizing society and economics have been tried on a major scale. These two things are not mutually exclusive. It could also be that crises are managed better now, but climate change still makes everything worse.

Another hint that climate overall is the dominant shaping factor can be found in our results around synchronization. For example, East African countries show the strongest negative correlations. This asynchrony likely reflects distinct regional climate drivers, particularly the Indian Ocean Dipole, which can produce rainfall patterns opposite to those affecting other major agricultural regions (Ummenhofer et al., 2009; Zheng et al., 2025).

The earlier food production shock study by Cottrell et al. (2019) also identified climate (and to a lesser extent conflict) as the main driver of disruption in food production. These two drivers may be causally linked. Zhang et al. (2011) showed how climate shocks reduce food production, which in turn triggers famine, conflict, and disease, ultimately leading to population decline. This means climate-driven crop failures can create the conditions for conflict. The prominence of both climate and conflict in our results fits with this pattern of cascading effects in food system disruptions.

All of this seems to apply especially to Europe, where many of the largest food shocks were caused by the 2003 heatwave alone. This, and the generally very high rate of climate-related shocks in Europe, highlight these regions as especially vulnerable to these kinds of shocks. However, European shocks are also often relatively small; this could be due to a more benign European climate or potentially because the agricultural systems there are better equipped to handle shocks.

The geographic patterns in shock magnitude we observe likely reflect not only differences in climate exposure and governance, but also regional crop composition. Southern Africa's extreme shocks occur in maize-dominated systems, where drought sensitivity is approximately twice that of wheat (Daryanto et al., 2016). Europe's wheat-based systems and Asia's flooded paddy rice systems show greater resilience to moderate water stress, though all crops remain vulnerable to severe drought. These crop-specific vulnerabilities interact with regional climate patterns to shape overall shock magnitudes.

**4.2 Large shocks can and do happen**

The results here confirm that very large crop production shocks happen quite regularly, with the median of the largest shocks being around 27%. However, this dramatically varies by region, with African countries experiencing the most extreme collapses (up to -80% in Southern Africa), while Asian and Central European nations typically face more moderate largest shocks (-5 to -15%). This is in contrast to shock frequency patterns — Cottrell et al. (2019) found that crop production losses occur most often in South Asian countries, for example. Global shocks are typically much smaller than this. This does not mean that they cannot reach similar magnitudes. Both the shape of the global shock distribution and our knowledge of history imply that such large shocks can also happen globally. For example, between the start and end of World War 2, global food availability per capita fell by something between 5% (FAO, 1955) to 12% (Collingham, 2012), though exact numbers are hard to come by and the effects were much worse in some locations. This global reduction consisted mostly of countries in Europe losing significant amounts of their production. Their losses often were around 20–40% (FAO, 1955), well within the range of the country-level shocks studied here. Data for World War 1 is much more scarce, but many European countries lost 40% and more of their food production and cut food rations by similar amounts (Offer, 1991). This implies that global shock to the food systems was likely in a similar range as World War 2.

All this means that future global shocks of 5% or more are both possible and plausible. Given the asynchronous nature of global food production, we seem to have some buffer against this. However, this buffer only works as long as the reason for the shock is not global. If there were an event that could hit all countries globally, or multiple distinct causes hitting different regions at the same time, there would be no buffer left. Also, the largest global shocks (e.g. a geomagnetic storm or high altitude electromagnetic pulses disrupting industry and thus agriculture (Moersdorf et al., 2024)) would likely be on top of the natural variability, meaning that if humanity got unlucky and a global shock hit in a year that already had a big share of large shocks, things would be even worse.

Our analysis also shows that climate causes both the most shocks and the most severe shocks. The cause here is mostly droughts, but there are also instances of significant disruptions due to cold spells. Several of the worst shocks that could affect agriculture globally also work via the climate. For example, nuclear winter could potentially decrease global land temperature by around 10°C (Coupe et al., 2019), leading to widespread disruption of food production (Xia et al., 2022). Another climate pathway, likely similar in its effects to nuclear winter, would be a large volcanic eruption (Cassidy and Mani, 2022). Finally, there is preliminary research that indicates that AMOC collapse could also lead to massive disruption of European climate and thus agriculture (Lenton et al., 2023).

**4.3 The role of trade**

Global food production is highly connected and very reliant on trade, with around a quarter of all food being traded internationally (Ji et al., 2024). While trade is generally helpful for food security, it also makes countries vulnerable to disruptions elsewhere (Wang et al., 2023). This is especially a problem in Europe, as it is mainly trading internally, while everybody shares the same climate (Keys et al., 2025). For the largest catastrophes (like large geomagnetic storms or a nuclear war), this could result in many countries losing most of their food imports (Jehn et al., 2024a). Recent modeling by Verschuur et al. (2024) demonstrates how compound 'polycrises' combining multiple shocks can overwhelm the food system's normal adaptive capacity, resulting in consumer price increases of 23–52% across all crops and affecting virtually all countries simultaneously. This shows how the buffering effect of trade becomes less effective during compound, global-scale disruptions.

This can become a problem for all those countries that are not able to produce enough food within their own borders. For example, Stehl et al. (2025) show that many countries are not able to produce the staples of their diet. Especially for starchy staples, those countries that are not able to produce enough on their own show a high overlap with those countries experiencing the largest crop production shocks shown in this study.

However, successful adaptation is possible with international cooperation. Kuhla et al. (2023) showed how the international community managed to limit wheat price spikes after Russia's invasion of Ukraine through brokered agreements allowing Black Sea exports and alternative European river routes, combined with fortunately high global harvests in 2022. However, it cannot be taken for granted that the global stocks will always be full or coordination will always be possible, as the Ukraine war only influenced a small fraction of global food production.

That being said, having sufficient production and trade alone does not necessarily mean that people have enough food to eat. At first glance, South Sudan's largest annual shock of 8.3% in 2017 appears relatively manageable in terms of food production. However, the withholding of food aid as a weapon of war led to a significant famine, with 100,000 facing starvation and over 40% of the country in urgent need of food aid (United Nations World Food Programme, 2017). Recent analysis by Bajaj et al. (2025) demonstrates that trade's stabilizing role varies systematically by income level, mitigating future climate impacts for 60% of low-income countries while aggravating impacts for 53% of high-income countries. Import-dependent lower-income countries often source from regions where climate change may increase production, whereas wealthier nations face amplified risks from climate impacts in their trading partners.

Even in the absence of direct conflict or trade complications, poor management can make food access much worse than any given yield shock. The Great Chinese Famine killed 16.5–45 million people between 1959 and 1961 despite average rural food availability being high enough to prevent severe famine (Meng et al., 2015). Excessive government procurement from rural farmers to urban areas, redirection of labour away from agriculture, and a plethora of other unfortunate policies led to a vast

number of unnecessary deaths (Kung and Lin, 2003). The key takeaway from these historical examples is that a future GCFF could lead to disastrous levels of famine if managed poorly, especially considering how difficult cooperation may be during a global crisis.

**4.4 Preparation is needed**

All this aims to highlight that our food system regularly experiences major shocks that can plausibly happen on a global scale as well. Governments should therefore take such major threats seriously and prepare accordingly. While we have global stocks of food, these usually only last for 0.5 to 1 year (Laio et al., 2016), meaning that they would not be enough for several-year shocks like large volcanic eruptions. Therefore, contingency plans are needed:

- Currently, very few national risk registers even grapple with global disruptions to the food system. For future risk assessments, such events should be included and planned for.

- Many of the shocks presented here also have the potential to influence each other through time, like a mismanagement in one year making a drought more difficult to cope with in the next. Future research could explore these interferences by tracking not only the reasons for the biggest shocks, but all detectable shocks.

- Trade partners should be diversified throughout different climate zones to enhance resilience (Keys et al., 2025). This is especially important, as in the current geopolitical climate, countries are reducing trade in general, while also preferentially trading with their closest allies (Piñeiro and Piñeiro, 2024). This diversification should also include countries that are both synchronous and asynchronous to global food production, e.g. trading with both Brazil and Germany. Similarly, a diversification of crops would also help, as different crops react differently to the same stressors. As Hertel et al. (2021) emphasize, diversification across crops, landscapes, income sources, and trade partners represents a fundamental strategy for building food system resilience at multiple scales. However, increased market integration can encourage production specialization even as it reduces overall risk exposure. Therefore, policies promoting resilience should consider how production, trade, and household diversification interact to avoid creating new vulnerabilities.

- Even after smaller, local food production shocks, countries quickly resort to export bans to ensure enough food for their citizens. These are often done much earlier than actually needed, leading to food insecurity, even if enough food is available globally (Puma et al., 2015). This means trade agreements between countries should explicitly plan out under what circumstances export bans would be considered.

- For some of the catastrophes that could affect the global food system, there is a need to build up alternative food sources to our present-day agriculture, which would be better suited to lower light/temperature or lower tech available. García Martínez et al. (2025) provide a systematic framework for resilient foods that could function under different catastrophic scenarios. This could include seaweed (Jehn et al., 2024b), mass-produced low-tech greenhouses (Alvarado et al., 2020), sugar from fiber (Throup et al., 2022), or protein from natural gas (García Martínez et al., 2022).

The dataset produced by this study opens several avenues for future research. First, tracking not only the largest shock but all detectable shocks for each country would reveal how sequential or compound events interact—for instance, whether mismanagement in one year amplifies vulnerability to drought the next. Second, and perhaps most policy-relevant, would be systematic case studies tracing each major shock from production loss through to human welfare outcomes. Key questions include: How did prices respond? Did trade partners maintain exports or impose bans? Which population groups bore the burden? What interventions (if any) mitigated impacts? Answering these questions would substantially improve our understanding of food system resilience and the conditions under which production shocks become humanitarian crises.

Ultimately, all of this (and likely more) is needed to make this world secure against large disruptions of food production. We should start now with preparation, as we still have time.

**Author contributions**

Conceptualization: FUJ, JM, NW

Data curation: FUJ

Formal analysis: FUJ, JM, LGG, SB

Funding acquisition: FUJ

Investigation: FUJ, JM

Methodology: FUJ, JM, LGG, SB

Project administration: FUJ

Software: FUJ, JM, LGG, SB

Supervision: FUJ

Validation: FUJ, SB

Visualization: FUJ, JM, LGG, SB

Writing—original draft: FUJ, JM

Writing—review & editing: FUJ, JM, LGG, SB, NW

**Data and Code Availability**

All code and data used for this study are available in the repository: https://github.com/allfed/Historical-Food-Shocks (Jehn and Mulhall, 2025)

**Acknowledgements**
We thank the Research Hub community and Zu-Grama for providing the funding that allowed us to conduct this research
project, and Yeshodhara Baskaran for facilitating the connection.

We are grateful to Daniel Hoyer, David Denkenberger, Michael Hinge and Juan B. García Martínez for helpful comments on
an earlier version of this manuscript.

We acknowledge the use of artificial intelligence (Claude 4 Sonnet) to assist with sentence-level text revision, coding support,
and accelerating literature searches to identify documented causes of the detected crop production shocks.
**Competing interests**
All authors declare that they do not have any competing interests.

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
