# Peer review of "The Largest Crop Production Shocks: Magnitude, Causes and Frequency"

_EGUsphere, 2025_

## Author Comment (AC2)

We would like to thank the reviewers and the editor for their helpful and constructive comments on the manuscript "The Largest Crop Production Shocks: Magnitude, Causes and Frequency".

We found all of the feedback to be useful in improving and sharpening the research. We have now updated the manuscript to address the comments. We believe it has been significantly strengthened and is now suitable for publication. Below we have listed the reviewer and editor comments in black, along with our responses in light green. Text that has been added to the manuscript is in *italics and darker green*.

**Reviewer #1**

**General comments**

This is an impressive and engaging analysis of global food production shocks and their causes. The study makes good use of FAO data to identify and interpret the largest national food production shocks (in total calorie terms) over 1961–2023. I found the paper clear, rich in narrative detail, and well organized. It provides a valuable dataset and synthesis that will be of interest to both researchers and policymakers. My comments below are intended to help the authors strengthen the framing, clarify assumptions, and ensure that key methodological choices are transparent and well justified.

We thank the reviewer for their very positive assessment of our paper.

**Specific comments**

This paper clearly builds on Cottrell et al. (2019). It would help readers if the introduction and discussion more explicitly distinguished this study's new contributions—for example through expanded data coverage, new cause-attribution methods, or additional insights.

We added an additional paragraph to the introduction to highlight the main connections and differences.

Our approach builds on previous work, such as Cottrell et al. (2019) and Anderson et al. (2023). However, rather than analyzing climate patterns that might cause shocks like Anderson et al. (2023) or identifying shocks across multiple food sectors like Cottrell et al. (2019), this paper systematically describes the worst crop production shock that each country experienced and why it happened. We believe this unique focus on the largest magnitude shocks highlights the greatest dangers that crop production faces, providing a comprehensive map of actual worst-case vulnerabilities rather than merely describing risk factors in general.

The assumption that all crops could be diverted to human consumption in a crisis is strong and may not hold in practice. Many of the listed crops (for example, seed cotton, maize, soy, and barley) are primarily non-food or feed crops. I encourage the authors to discuss this limitation in more detail or, if feasible, re-analyze using food-only fractions or "delivered calories" that account for feed conversion losses (see

Cassidy et al., 2013). Even a short sensitivity check would substantially strengthen the robustness of the findings. Otherwise your findings would be weighted heavily toward feed crops and those countries that produce them (e.g., consider that only about a quarter of produced calories from maize ends up as human calories).

We discuss our reasoning now in more detail and have also done a sensitivity analysis, which shows that this does not lead to a skew in the results, regardless:

We do not differentiate between which of these crops are intended for feed or food, because in a famine situation, we assume that most, if not all, of it would be used for human consumption. We recognize that this does not reflect current food consumption patterns, because several of the crops (like maize or soya beans) are mostly used for feed and only 55 % of global crop calories reach humans directly (Cassidy et al., 2013). However, our aim is to quantify crop production shocks, rather than current consumption patterns. During severe food crises, feed is often redirected towards human consumption. For example, there are documented cases of this phenomenon for both World Wars (Collingham, 2012; Offer, 1991) and during the Great Chinese Famine (Meng et al., 2015). Depending on the crop, this might take some time and infrastructure, but it represents a sensible crisis response. Most of the crops we consider here are directly edible by humans. The crops used here, which are likely the most difficult for humans to consume, are seed cotton, rapeseed, and soya beans. To assess whether this changes our findings, we redid the analysis excluding seed cotton, rapeseed and soya beans. The results stay almost exactly the same, and for most countries, the results only change by a percentage point or less. This can also be seen in Figure S1, which is a version of Figure 2 but without those crops. The changes are so small that they are almost not detectable visually. We therefore conduct the analysis with the whole set of crops.

The Savitzky–Golay filter is well motivated, but it would be useful to show that results are not overly sensitive to this choice. I recommend testing one or two alternative detrending methods (e.g., Gaussian or LOESS) and examining whether the identified "largest shocks" or their magnitudes change materially. Similarly, reconsider the "must be below last year" rule or provide a robustness check without it.

We have added an explanation that highlights that the results between the two methods are rather small and provided a new supplementary figure to show it:

Though ultimately, a Gaussian filter and the Savitzky-Golay filter deliver very similar results for our dataset and identify similar magnitudes of shocks, as well as the same years with the largest shocks (Figure S2).

We also added an additional explanation to justify our usage of the additional constraint:

The additional constraint was added because the initial analysis incorrectly flagged years as shocks when yields had actually increased from the previous year. However, having more crops than the year before can hardly be considered a shock.

In your synchrony analysis, how do you account for the differences in size and production of different countries? Larger producers will naturally drive global totals, so weighting by production share or caloric contribution could yield a more accurate view of which regions most influence global variability. Clarifying or adding this weighting step would help the "buffering" interpretation.

We calculated correlations between each country and world production, excluding that country's contribution to avoid spurious correlations. Spearman correlation measures whether production changes move in the same direction simultaneously, rather than production magnitude—a country producing 2% or 20% of global calories shows identical correlation coefficients if their yields rise and fall synchronously.

Some of the observed geographic patterns may reflect differences in crop composition rather than exposure or governance. The authors could discuss this possibility, or test whether patterns persist when comparing regions growing similar crop mixes.

We agree that regional crop composition differences likely contribute to the geographic patterns in shock severity we observe and have added a paragraph to section 4.1 to address this:

The geographic patterns in shock magnitude we observe likely reflect not only differences in climate exposure and governance, but also regional crop composition. Extreme shocks in Southern Africa occur in maize-dominated systems, where drought sensitivity is approximately twice that of wheat (Lesk et al., 2022). Europe's wheat-based systems and Asia's flooded paddy rice systems show greater resilience to moderate water stress, though all crops remain vulnerable to severe drought. These crop-specific vulnerabilities interact with regional climate patterns to shape the overall shock magnitudes.

A few recent studies might offer useful methodological or interpretive ideas (and very sorry for the self citations—these do not need to be cited if not directly relevant):

- 1. On synchrony (sections 2.4, 3.4), see Mehrabi and Ramankutty (2019) and Egli et al. (2021). Drawing on their framework for quantifying and decomposing synchrony could strengthen your analysis.
- 2. On the role of trade (section 4.3), see Bajaj et al. (2025).
- 3. On diversification of trade (section 4.4), see Hertel et al. (2021).

Thank you for those helpful papers. We have extended the discussion:

For Section 4.3 (The role of trade):

Recent analysis by Bajaj et al. (2025) demonstrates that trade's stabilizing role varies systematically by income level, mitigating future climate impacts for 60% of low-income countries while aggravating impacts for 53% of high-income countries. Import-dependent lower-income countries often source from regions where climate change may increase production, whereas wealthier nations face amplified risks from climate impacts in their trading partners.

For Section 4.4 (Preparation is needed):

As Hertel et al. (2021) emphasize, diversification across crops, landscapes, income sources, and trade partners represents a fundamental strategy for building food system resilience at multiple scales. However, increased market integration can encourage production specialization even as it reduces overall risk exposure. Therefore, policies promoting resilience should consider how production, trade, and household diversification interact to avoid creating new vulnerabilities.

**Technical corrections**

1. Consider renaming section 3.2 to something like "Geographic patterns of shock types", because the geographic patterns of shocks has already been seen in figure 2 of section 3.1.

Changed as proposed.

2. Lines 221-222: The conclusion that "pests and diseases are not a major factor for the largest shocks" may be too strong given only one observed data point."

The fact that it is only one datapoint and that this datapoint is also rather small in comparison to the other categories hints that pests and diseases do not seem to be able to cause these major shocks. If they were more important, we would see more of them.

3. Lines 237-238: The reference to "North America" appears to describe tropical storm impacts in the Caribbean. "Central America" might be more appropriate, and the map (with only one blue country) suggests that this pattern could be toned down in the text.

Changed as proposed.

4. Lines 250-251: You discuss a decade of mismanagement under Idi Amin – it's not clear how a decade-long effect would result in a single year shock?

This depends somewhat on how we calculate shocks here. We do not calculate a year-to-year drop, but how much the production deviates from the long-term trend. Uganda had a rising production before Idi Amin took power and was able to recover afterwards. This means it had a high expected production, even in the years of Idi Amin. As the production got worse pretty much every year of his rule, this means the last year of his reign is detected as the largest shock.

5. Lines 274-275 & Figure 6: Because the 2020s decade includes only four years of data, the low count of shocks is expected. Normalizing by the number of years per decade (i.e., showing average shocks per year) would provide a fairer comparison.

We have chosen this way to visualize the figure intentionally. This way, we can more easily compare the actual values of shocks per decade. Additionally, although the 2020s have only four years of data in our comparison, this still indicates that they have fewer shocks than we initially expected. Even if we scale this by the ten years of the decade, we still end up with fewer shocks than every other decade. However, we have now noted this in the caption of Figure 6.

6. Lines 338-341: Please add supporting citations for these statements.

Changed as proposed.

7. Lines 342-347: I had a hard time understanding the argument in this paragraph. How does the Zhang et al. study (which only examined climate driven shocks, so did not compare it to other shocks) support the finding that climate is the main driver of shocks?

We have rephrased the paragraph to make the argument clearer:

The earlier food production shock study by Cottrell et al. (2019) also identified climate (and, to a lesser extent, conflict) as the main driver of disruptions in food production. These two drivers may be causally linked. Zhang et al. (2011) showed how climate shocks reduce food production, which in turn triggers famine, conflict, and disease, ultimately leading to population decline. This means climate-driven crop failures can create the conditions for conflict. The prominence of both climate and conflict in our results fits with this pattern of cascading effects in food system disruptions.

**Reviewer #2**

Nicely done study on crop production shocks, however I'm not sure if it brings so much new information to the table. The basic methodology of the study is analogous to Cottrell 2019 and Anderson 2023 which the authors already state. The added sophistication and differentiation in my view comes from the use of a

LLM to identify possible drivers of crop production shocks, as well as a different filter in the methodology

We added an additional paragraph to the introduction to highlight the main connections and differences.

Our approach here builds on previous work like Cottrell et al. (2019) and Anderson et al. (2023). However, rather than analyzing climate patterns that might cause shocks like Anderson et al. (2023) or identifying shocks across multiple food sectors like Cottrell et al. (2019), this paper systematically describes the worst crop production shock that each country experienced and why it happened. We think this unique focus on shocks of the largest magnitude highlights the greatest dangers crop production faces, providing a comprehensive map of actual worst-case vulnerabilities rather than describing risk factors in general.

(does the employment of a Gaussian filter change results much? Would be a relatively easy sensitivity test I imagine).

We have added an explanation that highlights that the results between the two methods are rather small and provided a new supplementary figure to show it:

Though ultimately, a Gaussian filter and the Savitzky-Golay filter deliver very similar results for our dataset and identify similar magnitudes of shocks, as well as the same years with the largest shocks (Figure S2).

The authors use FAOSTAT which provides more countries than Anderson, although less temporal extent; This is more useful for looking at strong shocks in individual countries, while globally synchronous shocks can already be mainly covered by a small number of countries. They also neglect the marine aspect which is already included in Cottrell, which in this paper's case, with its focus on individual

countries and large shocks, may be releavant as these are often island countries with low production so indeed marine sources of food could be interesting.

Marine food sources are an important part of the food system, and we do not question this. However, the paper is about crop production; therefore, marine sources fall outside the scope of our work. We do not include meat or animal products in the analysis for the same reason.

However, as the authors also note, the results are quite similar to what is already in the literature, that climatic factors, also ENSO are strong drivers of production shocks, along with geopolitical factors.

Our paper addresses a different research question than prior work. Previous studies characterized general patterns and frequencies of food production shocks. We instead ask: what are the absolute worst events that have occurred in each country's agricultural history, and what caused them?

This focus on extreme outliers reveals insights absent from the literature. We now know worst-case shocks vary dramatically by region (up to -80% in Africa versus -5% to -15% in Asia/Central Europe), that continent-level shocks above 5% happen every 1.8 years despite being rare globally, and that climate's role in the largest shocks has grown from 25% to 50-60% over time. The country-world correlation analysis, identifying which regions buffer versus amplify global shocks during extreme events, is also new.

When planning for catastrophic food failures, knowing the actual magnitude and causes of historical worst-case events provides an empirical foundation that general risk factor analysis cannot offer.

An added element here that could make the paper more interesting, also harnessing its integration of the LLM into the methodology, is to qualitatively trace the biophysical impacts back to human impacts - i.e. in years with production shocks were there reports of price inflation, shifts in global trade patterns, hunger indices, etc. This may be more possible now with the LLM doing the first screening.

We agree that this would make a fascinating research project. However, it is out of scope for this paper, as it would require a significant amount of work that is arguably larger than the paper presented here.

Finally, they note that some country-level data appears erroneous or unreliable. Can these be given an initial screen, or some way to account for reliability, especially the earlier FAOSTAT data is often quite dodgy.

As there is no objective cut-off on when to say that a specific time series is too unreliable, we rather include all of them here, not to arbitrarily exclude anything. Additionally, the unreliable data will just be added to the unknown causes category, as it would be impossible to find suitable literature. This means

| even if we exclude all unreliable data, it would not significantly change the results or the conclusions of the paper. |
|------------------------------------------------------------------------------------------------------------------------|
|                                                                                                                        |
|                                                                                                                        |
|                                                                                                                        |
|                                                                                                                        |
|                                                                                                                        |
|                                                                                                                        |
|                                                                                                                        |
|                                                                                                                        |
|                                                                                                                        |

---

## Author Response (AR2)

We would like to thank the reviewers and the editor for their helpful and constructive comments on the manuscript "The Largest Crop Production Shocks: Magnitude, Causes and Frequency ".

We found all of the feedback to be useful in improving and sharpening the research. We have now updated the manuscript to address the comments. We believe it has been significantly strengthened and is now suitable for publication. Below we have listed the reviewer and editor comments in black, along with our responses in light green. Text that has been added to the manuscript is in *italics and darker green*.

**Reviewer #1**

Thanks to the authors for their diligent revisions. I am satisfied with most of the revisions and/or responses to my comments from the first round. I have just a few more comments, mostly a conversational response, with some recommendations for minor revisions. Will be good to see your paper published.

We thank the reviewer for their positive assessment.

Comments

1.       I was originally still not entirely convinced by the argument that humans could eat crops intended for livestock feed, but granted I am not the expert on this. When I first made that comment, I was thinking of the #2 yellow corn predominantly grown in the US for livestock feed, which is a different variety from the sweet corn that people normally eat. But having done some more reading, it does seem like one could eat #2 corn in situations such as famine (e.g., https://tinyurl.com/47e85wk9; feel free to add this or other citations to make your argument stronger). I also tried to review the citations you added to support your argument that livestock feed has historically been diverted to human consumption during times of crises. Two of your citations (Collingham, Offer) are books I could not easily access. But I was able to quickly review the Meng et al. (2015) paper, which does say that peasants in rural China ate green crops (illegally) from the fields. So I accept this point. And besides, you have added a sensitivity analysis by removing cotton, soy and rapeseed.

Thank you for engaging with this point and for the additional reference on #2 corn during famine conditions. We appreciate that the Meng et al. (2015) evidence and our sensitivity analysis address your initial concern.

2.       I would suggest adding a short sentence (either to lines 213-215 or right after) saying that the spearman correlation also accounts for different production magnitudes of countries.

Changed as proposed.

3.       Regarding pests & diseases, I see your point. But there is a potentially important message here that you could touch upon in your discussion if you wish to. It is widely believed that pests and diseases are a major source of crop losses (e.g., see https://doi.org/10.1038/s41559-018-0793-y). This is in fact why farmers apply a lot of pesticides and why a lot of the GM crops are focussed on traits to reduce pest burden. But while the burden of pests and disease for crop production may be high, maybe it just remains at a high baseline without too much year-to-year fluctuation, and therefore does not create the types of episodic shocks that you are examining in this paper. In other words, some drivers are of high magnitude but without large fluctuations, and therefore we don't see their effects in terms of shocks.

We thank the reviewer for their helpful point here and added the following description to the discussion of pests and diseases:

*Pests and diseases are often one of the largest sources of crop losses (Savary et al., 2019). However, given that we do not find them here as one of the main causes of the largest shocks, this implies that they cause damage on a high magnitude but without large fluctuations.*

**Reviewer #2**

Thank you for the responses and the changes. I still find it hard difficult to distinguish novel fatures of this article as it stands. Of the three objectives listed in the introduction (L83) my impression is that these already exist in the literature, so this paper needs to argumentatively and/or technically assess a bit more.

I) I understand that the paper aims to focus on "the worst production shock" L78 and therefore aggregates, however, certain crops are particularly important for certain regions, and temporal dimensions also crucial i.e. alternating cropping seasons across hemispheres, as well as particular trade routes of particular crops. Furthermore, despite the justifcation on 'worst' shocks, the paper then goes on to describe production/shock distributions and coutry profiles, making the analysis extremely similar to Cottrell.

Furthermore, compared to Cottrell 2019 for instance, this paper is not only more aggregated, but actually covers less data, as Cottrell disaggregate between livestock, crops but also marine sources of food, which especially for small island states which this paper highlights crop shocks in, can be quite improtant. The famine framing allows this paper to assume all diversion of feed to direct consumption, but caloric content and calories available for human consumption can end up quite different (although I would disagree with soybean being one of the more difficult ones), and livestock may still play some roles in such cases, in any case, it is a strong simplification that already existing studies actually did not make. Especially as global disaster events could imaginably onset suddenly, adaptations of processing sectors may not happen very fast to make feed crops consumable to the general population.

We would like to thank the Reviewer for their perspective; however, we disagree that our paper does not bring novelty to the existing literature. The key distinction is that our study aggregates crop production by calories at the country level. In addition our focus is the most severe shocks to reflect the full historical spread of worst case calorie reduction. Calories rather than tonnes provide a more meaningful measure of food availability for human consumption, as calorie content differs widely between different foods and therefore covers actual human energy requirements better than tonnes.

Cottrell et al. analyze food shocks in general, while we are looking at the worst ones. It is important to note that the general distribution of shocks and the worst shocks are not caused by the exact same kind of events or concentrated in the same areas. In fact, we find clear differences such as Southern African countries experiencing the largest shocks in our analysis and South Asian countries experiencing the most frequent shocks in Cottrell et al. We have now noted this difference in section 4.2. Additionally, our analysis presents many insightful visualizations that were not present in Cottrell et al., like maps showing where to expect the largest shocks, a comparison of the magnitude of shocks, recurrence rates of shocks of different sizes and the synchronization analysis. In summary, Cottrell et al. describe **how often** food systems fluctuate; this paper asks **how bad it can get** when they fail. Therefore, we argue that both analyses, Cottrell et al and ours is a very worthwhile undertaking. We have outlined this explicitly now in our manuscript that these two studies complement each other.

*This study here complements Cottrell et al. (2019). While the earlier study focussed on how often the food system shows shocks in general, this study here explicitly focuses on how bad these shocks can get and why these most extreme shocks happen.*

In the first round of revisions, we have added a sensitivity analysis excluding crops most difficult to convert to human food (cotton, soy, rapeseed) showing robust results to our original analysis (see Figure S1). In addition, history has proven that feed-to-food conversion during crises has occurred several times and is well-documented during both World Wars and the Great Chinese Famine (already discussed in section 2.1)

We acknowledge that different crops vary in importance across regions. However, our analysis already includes all major crop types, which are aggregated consistently into a calorie-based measure of food availability. As a result, regionally dominant crops necessarily drive the aggregated calorie signal: a substantial decline in total calories can only occur if the crops that are most important in a given region experience a significant shock. In this sense, the aggregation preserves regional crop relevance rather than obscuring it.

This study is based on FAOSTAT data, which are available at annual temporal resolution. Consequently, we are unable to explicitly resolve seasonal cropping patterns or alternating growing seasons across hemispheres. While this is a limitation of the available data, annual aggregation remains appropriate for our focus on the worst historical calorie shocks, which by definition reflect cumulative production losses at the country level.

Lastly, we acknowledge that Cottrell et al. includes sources besides crops; however, our paper is explicitly focused on crop production shocks, as stated in the title. Also note that a large majority (82%, https://www.nature.com/articles/s43016-022-00573-0) of calories consumed by humans come directly from crops.

II) Regarding the synchrony analysis, how do the correlation coefficients change over time i.e. could analyse with a moving window, and see how the (a)synchrony that cannot be qualitatively ascribed to climate factors match with conflict or policy or other aspects? Or does synchrony change with different crops, northern hemisphere maize could be compensated by south american production for instance, but perhaps wheat is harder to do so?

Our analysis is meant to describe the synchronization across the whole time period. Nevertheless, we added an additional Figure S7 to showcase that there are changes in the synchronization over time, but that many of the broad patterns stay consistent.

Correlation of crop production changes between each country and the rest of the world

[Figure]

Spearman Correlation coefficient

*Figure S7: Figure 8: Correlation of crop production changes between each country and the rest of the world for the years 2003-2023. A positive correlation (blue) means a country's crop production tends to move in the same direction as global production. A negative correlation (red) means the country's production tends to move opposite to global trends.*

We have also extended the results section to mention this:

*These patterns of synchronization mostly stay consistent over time, but there are changes. For example, if we only look at the data from 2003 to 2023 (Figure S7), we can see that Brazil's relatively strong negative correlation becomes positive, while Europe's generally positive correlation and Africa's generally negative correlation persist. This likely reflects changes in agricultural practices and the dominance of certain regions when it comes to crop production.*

We further did the analysis for wheat only and also extended the results section for this:

[Figure]

*Figure S8: Figure 8: Correlation of only wheat production changes between each country and the rest of the world for the years 1961-2023. A positive correlation (blue) means a country's crop production tends to move in the same direction as global production. A negative correlation (red) means the country's production tends to move opposite to global trends. Light grey indicates no data.*

*Much of the synchronicity in global crop production is driven by wheat, which makes up a substantial share of total output (Figure S8). This means wheat's year-to-year variation can overshadow more localized patterns in other crops. For our purposes, this is not a limitation—we aim to understand global crop production as a whole and identify which regions might serve as buffers when others fail. The wheat-driven pattern shifts dominance toward major wheat exporters like Russia and Ukraine, whose production swings carry outsized weight in global totals. The United States, despite being a major wheat producer, shows weaker correlation with global trends. Possibly because its production variability operates independently of the factors driving Eurasian wheat yields, as both continents experience different climate impacts.*

III) Regarding the qualitative aspect, is there any indication that an AI-assisted search provides more systematic answers than a human-based one, and in which ways? Access to untranslated documents perhaps? but this needs to be validated, and/or mentioned, beyond the fact that it certainly saves time for the authhours.

While we do not claim an AI-assisted approach is superior to manual search per se, it is efficient and adequate for our purposes while, at the same time, allowing for higher coverage than humans could. Specifically, our validation workflow was as follows: Claude was used to surface potentially relevant documents, but *all* attributions were verified by human researchers reading the original sources. Sources returned include phrases such as "worst drought year since the mid-15th century" or "most violent and bloody period of the entire armed confrontation," suggesting our method successfully identifies extreme

events. For cases where AI did not surface any relevant information, we conducted additional manual searches. This methodology is described in detail in section 2.2.

Finally, I don't necessarily think closing the loop from driver to shock to (qualitative)human impact is beyond the scope of this analysis, especially given the framing of this paper on 'worst', human disaster, catastrophe etc, and how it aims to 'ground' such framings in historical data. I think it's extremely pertinent to then see if, given the 'worst' shock a country has faced in quantified history, how human impacts appeared, i.e. via price shocks, distributional impacts (on producers vs urban poor for instance), shifts in consumption, etc? And if these did not appear, what mechanisms allowed countries to avoid such impacts, i.e. was storage or trade particularly important under certain policy constraints. This would also allow for providing specific understandings of possible recommendations beyond the generalized ones given in section 4.4, would strenghten the framing and novelty of the paper much more beyond repeating an existing analysis.

We agree that tracing pathways from production shocks through price effects, distributional impacts, and consumption shifts would be valuable. However, this would constitute a substantial research programme in its own right— requiring detailed case studies for each of the country-level shocks identified. This is beyond the scope of a revision and represents a direction for future work, which we now note in Section 4.4:

*The dataset produced by this study opens several avenues for future research. First, tracking not only the largest shock but all detectable shocks for each country would reveal how sequential or compound events interact—for instance, whether mismanagement in one year amplifies vulnerability to drought the next. Second, and perhaps most policy-relevant, would be systematic case studies tracing each major shock from production loss through to human welfare outcomes. Key questions include: How did prices respond? Did trade partners maintain exports or impose bans? Which population groups bore the burden? What interventions (if any) mitigated impacts? Answering these questions would substantially improve our understanding of food system resilience and the conditions under which production shocks become humanitarian crises.*

Our paper's contribution is to systematically identify the largest shocks and explain why they occurred. Understanding how societies responded to each shock is an important but distinct research question.